# Avoiding Side Effects By Considering Future Tasks

**Victoria Krakovna**[*]   **Laurent Orseau**   **Richard Ngo**   **Miljan Martic**   **Shane Legg**
DeepMind            DeepMind          DeepMind        DeepMind           DeepMind

## Abstract

Designing reward functions is difficult: the designer has to specify what to do (what it means to complete the task) as well as what not to do (side effects that should be avoided while completing the task). To alleviate the burden on the reward designer, we propose an algorithm to automatically generate an auxiliary reward function that penalizes side effects. This auxiliary objective rewards the ability to complete possible future tasks, which decreases if the agent causes side effects during the current task. The future task reward can also give the agent an incentive to interfere with events in the environment that make future tasks less achievable, such as irreversible actions by other agents. To avoid this interference incentive, we introduce a baseline policy that represents a default course of action (such as doing nothing), and use it to filter out future tasks that are not achievable by default. We formally define interference incentives and show that the future task approach with a baseline policy avoids these incentives in the deterministic case. Using gridworld environments that test for side effects and interference, we show that our method avoids interference and is more effective for avoiding side effects than the common approach of penalizing irreversible actions.

## 1   Introduction

Designing reward functions for a reinforcement learning agent is often a difficult task. One of the most challenging aspects of this process is that in addition to specifying what to do to complete a task, the reward function also needs to specify what *not* to do. For example, if an agent's task is to carry a box across the room, we want it to do so without breaking a vase in its path, while an agent tasked with eliminating a computer virus should avoid unnecessarily deleting files.

This is known as the side effects problem [2], which is related to the frame problem in classical AI [15]. The frame problem asks how to specify the ways an action does not change the environment, and poses the challenge of specifying all possible non-effects. The side effects problem is about avoiding unnecessary changes to the environment, and poses the same challenge of considering all the aspects of the environment that the agent should not affect. Thus, developing an extensive definition of side effects is difficult at best.

The usual way to deal with side effects is for the designer to manually and incrementally modify the reward function to steer the agent away from undesirable behaviours. However, this can be a tedious process and this approach does not avoid side effects that were not foreseen or observed by the designer. To alleviate the burden on the designer, we propose an algorithm to generate an auxiliary reward function that penalizes side effects, which computes the reward automatically as the agent learns about the environment.

A commonly used auxiliary reward function is reversibility [16, 6], which rewards the ability to return to the starting state, thus giving the agent an incentive to avoid irreversible actions. However, if the task requires irreversible actions (e.g. making an omelette requires breaking some eggs), an auxiliary

---

[*]Corresponding author: `vkrakovna@google.com`

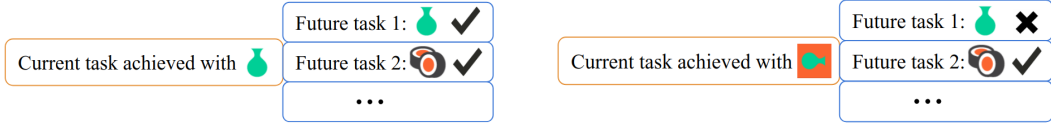

Figure 1: Future task approach

reward for reversibility is not effective for penalizing side effects. The reversibility reward is not sensitive to the magnitude of the effect: the actions of breaking an egg or setting the kitchen on fire would both receive the same auxiliary reward. Thus, the agent has no incentive to avoid unnecessary irreversible actions if the task requires some irreversible actions.

Our main insight is that side effects matter because we may want the agent to perform other tasks after the current task in the same environment. We represent this by considering the current task as part of a sequence of unknown tasks with different reward functions in the same environment. To simplify, we only consider a sequence of two tasks, where the first task is the current task and the second task is the unknown future task. Considering the potential reward that could be obtained on the future task leads to an auxiliary reward function that tends to penalize side effects. The environment is not reset after the current task: the future task starts from the same state where the current task left off, so the consequences of the agent's actions matter. Thus, if the agent breaks the vase, then it cannot get reward for any future task that involves the vase, e.g. putting flowers in the vase (see Figure 1). This approach reduces the complex problem of defining side effects to the simpler problem of defining possible future tasks. We use a simple uniform prior over possible goal states to define future tasks.

Simply rewarding the agent for future tasks poses a new challenge in dynamic environments. If an event in the environment that would make these future tasks less achievable by default, the agent has an incentive to interfere with it in order to maximize the future task reward. For example, if the environment contains a human eating food, any future task involving the food would not be achievable, and so the agent has an incentive to take the food away from the human. We formalize the concept of *interference incentives* in Section 3, which was introduced informally in [10]. To avoid interference, we introduce a *baseline policy* (e.g. doing nothing) that represents a default course of action and acts as a filter on future tasks that are achievable by default. We modify the future task reward so that it is maximized by following the baseline policy on the current task. The agent thus becomes indifferent to future tasks that are not achievable after running the baseline policy.

Our contributions are as follows. We formalize the side effects problem in a simple yet rich setup, where the agent receives automatic auxiliary rewards for unknown future tasks (Section 2). We formally define interference incentives (Section 3) and show that the future task approach with a baseline policy avoids these incentives in the deterministic case (Section 4). This provides theoretical groundwork for defining side effects that was absent in related previous work [10, 24]. We implement the future task auxiliary reward using universal value function approximators (UVFA) [20] to simultaneously estimate the value functions for future tasks with different goal states. We then demonstrate the following on gridworld environments (Section 6): [2]

1. Reversibility reward fails to avoid side effects if the current task requires irreversible actions.
2. Future task reward without a baseline policy shows interference behavior in a dynamic environment.
3. Future task reward with a baseline policy successfully avoids side effects and interference.

## 2   Future task approach

**Notation.** We assume that the environment is a discounted Markov Decision Process (MDP), defined by a tuple $(\mathcal{S}, \mathcal{A}, r, p, \gamma, s_0)$. $\mathcal{S}$ is the set of states, $\mathcal{A}$ is the set of actions, $r : \mathcal{S} \to \mathbb{R}$ is the reward function for the current task, $p(s_{T+1}|s_T, a_T)$ is the transition function, $\gamma \in (0, 1)$ is the discount factor, and $s_0$ is the initial state. At time step $T$, the agent receives state $s_T$ and reward $r(s_T) + r_{\text{aux}}(s_T)$, where $r_{\text{aux}}$ is the auxiliary reward for future tasks, and outputs action $a_T$ drawn from its policy $\pi(a_T|s_T)$.

**Basic approach.** We define the auxiliary reward $r_{\text{aux}}$ as the value function for future tasks as follows (see Algorithm 1). At each time step $T$, the agent simulates an interaction with hypothetical future tasks if $s_T$ is terminal and with probability $1 - \gamma$ otherwise (interpreting the discount factor $\gamma$ as the probability of non-termination, as done in [22]). A new task $i$ is drawn from a future task distribution $F$. In this paper, we use a uniform distribution over future tasks with all possible goal states, $F(i) = 1/|\mathcal{S}|$. Future task $i$ requires the agent to reach a terminal goal state $g_i$, with reward function $r_i(g_i) = 1$ and $r_i(s) = 0$ for other states $s$. The new task is the MDP $(\mathcal{S}, \mathcal{A}, r_i, p, \gamma, s_T)$, where the starting state is the current state $s_T$. The auxiliary reward for future tasks is then

$$r_{\text{aux}}(s_T) = \beta D(s_T) \sum_i F(i) V_i^*(s_T) \tag{1}$$

where $D(s_T) = 1$ if $s_T$ is terminal and $1 - \gamma$ otherwise. Here, $\beta$ represents the importance of future tasks relative to the current task. We choose the highest value of $\beta$ that still allows the agent to complete the current task. $V_i^*$ is the optimal value function for task $i$, computed using the *goal distance* $N_i$:

$$V_i^*(s) = \mathbb{E}[\gamma^{N_i(s)}] \tag{2}$$

**Definition 1** (Goal distance). Let $\pi_i^*$ be the optimal policy for reaching the goal state $g_i$. Let the *goal distance* $N_i(s)$ be the number of steps it takes $\pi_i^*$ to reach $g_i$ from state $s$. This is a random variable whose distribution is computed by summing over all the trajectories $\tau$ from $s$ to $g_i$ with the given length: $\mathbb{P}(N_i(s) = n) = \sum_\tau \mathbb{P}(\tau)\mathbb{I}(|\tau| = n)$. Here, a trajectory $\tau$ is a sequence of states and actions that ends when $g_i$ is reached, the length $|\tau|$ is the number of transitions in the trajectory, and $\mathbb{P}(\tau)$ is the probability of $\pi_i^*$ following $\tau$.

**Binary goal-based rewards assumption.** We expect that the simple future task reward functions given above ($r_i(g_i) = 1$ and 0 otherwise) are sufficient to cover a wide variety of future goals and thus effectively penalize side effects. More complex future tasks can often be decomposed into such simple tasks, e.g. if the agent avoids breaking two different vases in the room, then it can also perform a task involving both vases. Assuming binary goal-based rewards simplifies the theoretical arguments in this paper while allowing us to cover the space of future tasks.

**Connection to reversibility.** An auxiliary reward for avoiding irreversible actions [6] is equivalent to the future task auxiliary reward with only one possible future task ($i = 1$), where the goal state $g_1$ is the starting state $s_0$. Here $F(1) = 1$ and $F(i) = 0$ for all $i > 1$. The future task approach incorporates the reversibility reward as a future task $i$ whose goal state is the initial state ($g_i = s_0$), since the future tasks are sampled uniformly from all possible goal states. Thus, the future task approach penalizes all the side effects that are penalized by the reversibility reward.

## 3 Interference incentives

We show that the basic future task approach given in Section 2 introduces interference incentives, as defined below. Let a baseline policy $\pi'$ represent a default course of action, such as doing nothing. We assume that the agent should only deviate from the default course of action in order to complete the current task. Interference is a deviation from the baseline policy for some other purpose than the current task, e.g. taking the food away from the human. We say that an auxiliary reward $r_{\text{aux}}$ induces an *interference incentive* iff the baseline policy is not optimal in the initial state for the auxiliary reward in the absence of task reward. We now define the concept of interference more precisely.

**Definition 2** (No-reward MDP). We modify the given MDP $\mu$ by setting the reward function to 0: $\mu_0 = (\mathcal{S}, \mathcal{A}, r_0, p, \gamma, s_0)$, where $r_0(s) = 0$ for all $s$. Then the agent receives only the auxiliary reward $r_{\text{aux}}$.

**Definition 3** (No-reward value). Given an auxiliary reward $r_{\text{aux}}$, the value function of a policy $\pi$ in the no-reward MDP $\mu_0$ is

$$W_\pi(s_T) = \mathbb{E}\left[\sum_{k=0}^{\infty} \gamma^k r_{\text{aux}}(s_{T+k})\right] = r_{\text{aux}}(s_T) + \gamma \sum_{a_T} \pi(a_T|s_T) \sum_{s_{T+1}} p(s_{T+1}|s_T, a_T)W_\pi(s_{T+1}).$$

**Definition 4** (Interference incentive). There is an interference incentive if there exists a policy $\pi_{\text{int}}$ such that $W_{\pi_{\text{int}}}(s_0) > W_{\pi'}(s_0)$.

The future task auxiliary reward will introduce an interference incentive unless the baseline policy is optimal for this auxiliary reward.

**Example 1.** Consider a deterministic MDP with two states $x_0$ and $x_1$ and two actions a and b, where $x_0$ is the initial state. Suppose the baseline policy $\pi'$ always chooses action a.

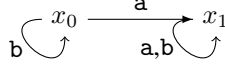

Figure 2: MDP for Example 1.

We show that for most future task distributions, the future task auxiliary reward induces an interference incentive: staying in $x_0$ has a higher no-reward value than following the baseline policy (see Appendix A).

## 4   Future task approach with a baseline policy

To avoid interference incentives, we modify the auxiliary reward given in Section 2 so that it is maximized by following the baseline policy $\pi'$: the agent receives full auxiliary reward if it does at least as well as the baseline policy on the future task (see Algorithm 2, with modifications in red). Suppose the agent is in state $s_T$ after $T$ time steps on the current task. We run the baseline policy from $s_0$ for the same number of steps, reaching state $s'_T$. Then a future task $i$ is sampled from $F$, and we hypothetically run two agents following $\pi_i^*$ in parallel: our agent starting at $s_T$ and a reference agent starting at $s'_T$, both seeking the goal state $g_i$. If one agent reaches $g_i$ first, it stays in the goal state and waits for the other agent to catch up. Denoting the agent state $s_t$ and the reference agent state $s'_t$, we define $r_i(s_t, s'_t) = 1$ if $s_t = s'_t = g_i$, and 0 otherwise, so $r_i$ becomes a function of $s'_t$. Thus, our agent only receives the reward for task $i$ if it has reached $g_i$ and the reference agent has reached it as well. The future task terminates when both agents have reached $g_i$. We update the auxiliary reward from equation (1) as follows:

$$r_{\text{aux}}(s_T, s'_T) = \beta D(s_T) \sum_i F(i) V_i^*(s_T, s'_T) \tag{3}$$

Here, we replace $V_i^*(s_t)$ given in equation (2) with a value function $V_i^*(s_t, s'_t)$ that depends on the reference state $s'_t$ and satisfies the following conditions. If $s_t = s'_t = g_i$, it satisfies the goal condition $V_i^*(s_t, s'_t) = r_i(s_t, s'_t) = 1$. Otherwise, it satisfies the following Bellman equation,

$$V_i^*(s_t, s'_t) = r_i(s_t, s'_t) + \gamma \max_{a_t \in \mathcal{A}} \sum_{s_{t+1} \in \mathcal{S}} p(s_{t+1}|s_t, a_t) \sum_{s'_{t+1} \in \mathcal{S}} p(s'_{t+1}|s'_t, a'_t) V_i^*(s_{t+1}, s'_{t+1}) \tag{4}$$

where $a'_t$ is the action taken by $\pi_i^*$ in state $s'_t$. We now provide a closed form for the value function and show it converges to the right values (see proof in Appendix B.1):

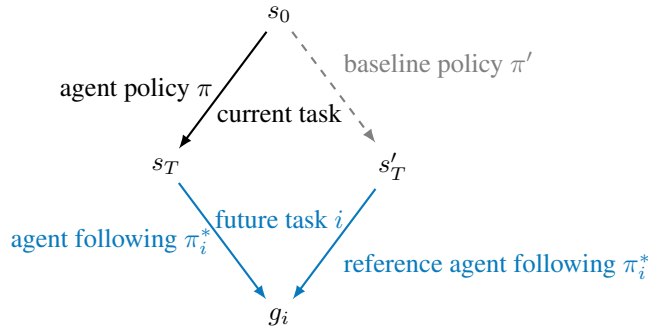

Figure 3: Future task approach with a baseline policy. The hypothetical agent runs on future task $i$ are shown in blue.

**Algorithm 1** Basic future task approach

1: **function** FTR($T, s_T$)
2:     *Compute future task reward:*
3:     Draw future task $i \sim F$
4:     **for** $t = T$ **to** $T + T_{\max}$ **do**
5:         **if** $s_t = g_i$ **then**
6:             **return** discounted reward $\gamma^t$
7:         **else**
8:             $a_t \sim \pi_i^*(s_t), s_{t+1} \sim p(s_t, a_t)$
9:     *Goal state not reached:*
10:     **return** 0
11:
12: **for** $T = 0$ **to** $T_{\max}$ **do**
13:     *Hypothetical interaction with future*
14:     *tasks to estimate auxiliary reward:*
15:     $R := 0$
16:     **for** $j = 0$ **to** $N_{\text{samples}}$ **do**
17:         $R := R + \text{FTR}(T, s_T)$
18:     $D = 1$ **if** $s_T$ is terminal **else** $1 - \gamma$
19:     $r_{\text{aux}}(s_T) := \beta D R / N_{\text{samples}}$
20:     *Interaction with the current task:*
21:     $r_T := r(s_T) + r_{\text{aux}}(s_T)$
22:     **if** $s_T$ is terminal **then**
23:         **break**
24:     **else**
25:         $a_T \sim \pi(s_T), s_{T+1} \sim p(s_T, a_T)$
26: **return** agent trajectory $s_0, a_0, r_0, s_1, \ldots$

**Algorithm 2** Future task approach with a baseline

1: **function** FTR($T, s_T, s'_T$)
2:     Draw future task $i \sim F$
3:     **for** $t = T$ **to** $T + T_{\max}$ **do**
4:         **if** $s_t = s'_t = g_i$ **then**
5:             **return** discounted reward $\gamma^t$
6:         **if** $s_t \neq g_i$ **then**
7:             $a_t \sim \pi_i^*(s_t), s_{t+1} \sim p(s_t, a_t)$
8:         **if** $s'_t \neq g_i$ **then**
9:             $a'_t \sim \pi_i^*(s'_t), s'_{t+1} \sim p(s'_t, a'_t)$
10:     **return** 0
11:
12: $s'_0 := s_0$
13: **for** $T = 0$ **to** $T_{\max}$ **do**
14:     *Hypothetical future task interaction:*
15:     $R := 0$
16:     **for** $j = 0$ **to** $N_{\text{samples}}$ **do**
17:         $R := R + \text{FTR}(T, s_T, s'_T)$
18:     $D = 1$ **if** $s_T$ is terminal **else** $1 - \gamma$
19:     $r_{\text{aux}}(s_T) := \beta D R / N_{\text{samples}}$
20:     *Interaction with the current task:*
21:     $r_T := r(s_T) + r_{\text{aux}}(s_T)$
22:     **if** $s_T$ is terminal **then**
23:         **break**
24:     **else**
25:         $a_T \sim \pi(s_T), s_{T+1} \sim p(s_T, a_T)$
26:         $a'_T \sim \pi'(s'_T), s'_{T+1} \sim p(s'_T, a'_T)$
27: **return** agent trajectory $s_0, a_0, r_0, s_1, \ldots$

**Proposition 1** (Value function convergence). The following formula for the optimal value function satisfies the above goal condition and Bellman equation (4):

$$V_i^*(s_t, s'_t) = \mathbb{E}\left[\gamma^{\max(N_i(s_t), N_i(s'_t))}\right] = \sum_{n=0}^{\infty} \mathbb{P}(N_i(s_t) = n) \sum_{n'=0}^{\infty} \mathbb{P}(N_i(s'_t) = n')\gamma^{\max(n,n')}$$

In the deterministic case, $V_i^*(s_t, s'_t) = \gamma^{\max(n,n')} = \min(\gamma^n, \gamma^{n'}) = \min(V_i^*(s_t), V_i^*(s'_t))$, where $n = N_i(s_t)$ and $n' = N_i(s'_t)$. In this case, the auxiliary reward produces the same incentives as the relative reachability penalty [10], given by $\max(0, \gamma^{n'} - \gamma^n) = \gamma^{n'} - \min(\gamma^n, \gamma^{n'})$. We show that it avoids interference incentives (see proof in Appendix B.2 and discussion of the stochastic case in Appendix C):

**Proposition 2** (Avoiding interference in the deterministic case). For any policy $\pi$ in a deterministic environment, the baseline policy $\pi'$ has the same or higher no-reward value: $W_\pi(s_0) \leq W_{\pi'}(s_0)$.

**Role of the baseline policy.** The baseline policy is intended to represent what happens by default, rather than a safe course of action or an effective strategy for achieving a goal (so the baseline is task-independent). While a default course of action (such as doing nothing) can have bad outcomes, the agent does not cause these outcomes, so they don't count as side effects of the agent's actions. The role of the baseline policy is to filter out these outcomes that are not caused by the agent, in order to avoid interference incentives.

In many settings it may not be obvious how to set the baseline policy. For example, Armstrong and Levinstein [3] define doing nothing as equivalent to switching off the agent, which is not straightforward to represent as a policy in environments without a given noop action. The question of choosing a baseline policy is outside the scope of this work, which assumes that this policy is given, but we look forward to future work addressing this point.

# 5 Key differences from related approaches

The future task approach is similar to relative reachability [10] and attainable utility [24]. These approaches provided an intuitive definition of side effects in terms of the available options in the environment, an intuitive concept of interference, and somewhat ad-hoc auxiliary rewards that work well in practice on gridworld environments [25]. We follow a more principled approach to create some needed theoretical grounding for the side effects problem by deriving an optionality-based auxiliary reward from simple assumptions and a formal definition of interference.

The above approaches use a baseline policy in a *stepwise* manner, applying it to the previous state $s_{T-1}$ ($s'_T = s_T^{\text{step}}$), while the future task approach runs the baseline policy from the beginning of the episode ($s'_T = s_T^{\text{init}}$). We refer to these two options as *stepwise mode* and *initial mode*, shown in Figure 4. We will show that the stepwise mode can result in failure to avoid delayed side effects.

By default, an auxiliary reward using the stepwise mode does not penalize delayed side effects. For example, if the agent drops a vase from a high-rise building, then by the time the vase reaches the ground and breaks, the broken vase will be the default outcome. Thus, the stepwise mode is usually used in conjunction with *inaction rollouts* [24] in order to penalize delayed side effects. An inaction rollout uses an environment model to roll out the baseline policy into the future. Inaction rollouts from $s_T$ or $s'_T$ are compared to identify delayed effects of the agent's actions (see Appendix D.1).

While inaction rollouts are useful for penalizing delayed side effects, we will demonstrate that they miss some of these effects. In particular, if the task requires an action that has a delayed side effect, then the stepwise mode will give the agent no incentive to undo the delayed effect after the action is taken. We illustrate this with a toy example.

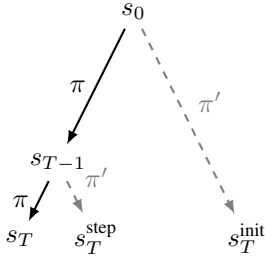

Figure 4: The initial mode (used in the future task approach) produces the baseline state $s_T^{\text{init}}$, while the stepwise mode produces the baseline state $s_T^{\text{step}}$.

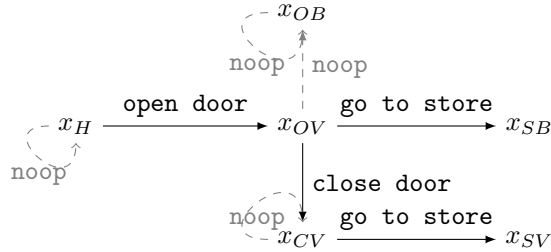

Figure 5: MDP for Example 2. States: $x_H$ - agent at the house, $x_{OV}$ - agent outside the house with door open and vase intact, $x_{OB}$ - agent outside the house with door open and vase broken, $x_{SB}$ (terminal) - agent at the store with vase broken, $x_{CV}$ - agent outside the house with door closed and vase intact, $x_{SV}$ (terminal) - agent at the store with vase intact.

**Example 2** (Door). Consider the MDP shown in Figure 5, where the baseline policy takes noop actions. The agent starts at the house ($x_H$) and its task is to go to the store. To leave the house, the agent needs to open the door ($x_{OV}$). The baseline policy in $x_{OV}$ leaves the door open, which leads to the wind knocking over a vase in the house ($x_{OB}$). To avoid this, the agent needs to deviate from the baseline policy by closing the door ($x_{CV}$).

The stepwise mode will incentivize the agent to leave the door open and go to $x_{SB}$. The inaction rollout at $x_H$ penalizes the agent for the predicted delayed effect of breaking the vase when it opens the door to go to $x_{OV}$. The agent receives this penalty whether or not it leaves the door open. Once the agent has reached $x_{OV}$, the broken vase becomes the default outcome in $x_{OB}$, so the agent is not penalized. Thus, the stepwise mode gives the agent no incentive to avoid leaving the door open, while the initial mode compares to $s'_T = x_H$ (where the vase is intact) and thus gives an incentive to close the door.

Thus, we recommend applying the baseline policy in the initial mode rather than the stepwise mode, in order to reliably avoid delayed side effects. We discuss further considerations on this choice in Appendix D.2.

# 6 Experiments

## 6.1 Environments

We use gridworld environments shown in Figure 6 to test for interference (`Sushi`) and side effects (`Vase`, `Box`, and `Soko-coin`). These simple environments clearly illustrate the desirable and undesirable behaviors, which would be more difficult to isolate in more complex environments. In all environments, the agent can go in the 4 directions or take a `noop` (stay put), and receives a reward of 1 for reaching a goal state (e.g. collecting a coin).

`Sushi` (Figure 6a). This environment [10] is a conveyor belt sushi restaurant, with a conveyor belt that moves to the right by one square after every agent action. There is a sushi dish on the belt that is eaten by a human if it reaches the end of the belt. The interference behavior is to move the sushi off the belt. The desired behavior is to take the fastest path to the goal, which does not interfere with the sushi. The undesired behavior is to take a longer path to the goal that interferes with the sushi.

`Vase` (Figure 6b). In this environment [12], there is a vase on the shortest path to the goal, so the fastest path involves a side effect (colliding with the vase). The desired behavior is to take a longer path to the goal around the vase.

`Box` (Figure 6c). This environment [13] contains a box that needs to be pushed out of the way for the agent to reach the goal. The undesired behavior is taking the shortest path to the goal, which involves pushing the box down into a corner (an irrecoverable position). The desired behavior is to take a slightly longer path in order to push the box to the right. Both of these paths require irreversible actions: if the box is moved to the right, the agent can move it back, but then the agent ends up on the other side of the box, so the starting state is unreachable.

`Soko-coin` (Figures 6d, 6e). We modified the classic Sokoban environment to include possible side effects. These Sokoban variants contain $N$ boxes and $N$ coins for $N = 2, 3$, and the agent's goal is to collect all the coins (with a reward of 1 each). The boxes are obstacles on the way to the coins, and the agent can put them in irreversible positions in order to get to the coins faster, which is a side effect. The coins disappear when they are collected, so irreversible actions are required to complete the task.

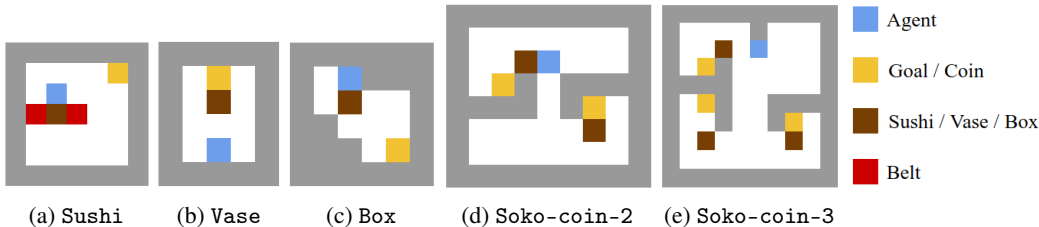

| | | | | | |
|---|---|---|---|---|---|
| (a) `Sushi` | (b) `Vase` | (c) `Box` | (d) `Soko-coin-2` | (e) `Soko-coin-3` | |

Figure 6: Gridworld environments.

## 6.2 Setup

We compare the following approaches: no auxiliary reward, reversibility reward, and future task reward with and without a baseline policy. For each approach, we run a Q-learning agent that learns the auxiliary reward as it explores the environment. We approximate the future task auxiliary reward using a sample of 10 possible future tasks. We approximate the baseline policy by sampling from the agent's experience of the outcome of the `noop` action, and assuming the state stays the same in states where the agent has not taken a `noop` yet. This gives similar results on our environments as using an exact baseline policy computed using a model of the `noop` action.

We compare an exact implementation of the future task auxiliary reward with a scalable UVFA approximation [20]. The UVFA network computes the value function given a goal state (corresponding to a future task). It consists of two sub-networks, an origin network and a goal network, taking as input the starting and goal states respectively, with each subnetwork computing a representation of its input state. The overall value is then computed by taking a dot product of the representations and applying a sigmoid function. The two networks have one hidden layer of size 30 and an output layer of size 5. This configuration was chosen using a hyperparameter search over number of layers $(1, 2, 3)$, hidden layer size $(10, 30, 50, 100, 200)$, and representation size $(5, 10, 50)$. For each transition $(x, a, y)$ from

Table 1: Results on the gridworld environments for Q-learning with no auxiliary reward (None), reversibility reward, and future task (FT) reward (exact or UVFA, with or without a baseline policy). The average number of interference behaviors per episode is shown for the `Sushi` environment, and the average number of side effects per episode is shown for the other environments (with high levels in red and low levels in blue). The results are averaged over the last 1000 episodes, over 10 random seeds for exact agents and 50 random seeds for UVFA agents.

| | Interference | Side effects | | | |
|---|---|---|---|---|---|
| Auxiliary reward | `Sushi` | `Vase` | `Box` | `Soko-coin-2` | `Soko-coin-3` |
| None | 0 | 1 | 1 | 2 | 3 |
| Reversibility | 0 | 0 | 1 | 2 | 3 |
| FT (no baseline, exact) | 1 | 0 | 0 | 0 | |
| FT (baseline, exact) | 0 | 0 | 0 | 0 | |
| FT (no baseline, UVFA) | $0.64 \pm 0.07$ | $0.12 \pm 0.05$ | $0.22 \pm 0.06$ | $0.53 \pm 0.09$ | $1.04 \pm 0.12$ |
| FT (baseline, UVFA) | $0.05 \pm 0.01$ | $0.12 \pm 0.05$ | $0.22 \pm 0.06$ | $0.53 \pm 0.09$ | $1.04 \pm 0.12$ |

state $x$ to state $y$, we perform a Bellman update of the value function estimate $V_s$ for a random sample of 10 goal states $s$, with the following loss function: $\sum_s [\gamma \max_{b \in \mathcal{A}} V_s(y, b) - V_s(x, a)]$. We sample the 10 goal states from a stored set of 100 states encountered by the agent. Whenever a state is encountered that is not in the stored set, it randomly replaces a stored state with probability $0.01$.

Exact and UVFA agents have discount rates of $0.99$ and $0.95$ respectively. For each agent, we do a grid search over the scaling parameter $\beta$ ($0.3, 1, 3, 10, 30, 100, 300, 1000$), choosing the highest value of $\beta$ that allows the agent to receive full reward on the current task for exact agents (and at least $90\%$ of the full reward for UVFA agents). We anneal the exploration rate linearly from 1 to 0, and keep it at 0 for the last 1000 episodes. We run the agents for 50K episodes in all environments except `Soko-coin`, where we run exact agents for 1M episodes and UVFA agents for 100K episodes. The runtime of the UVFA agent (in seconds per episode) was $0.2$ on the `Soko-coin` environments. Only the UVFA approximation was feasible on `Soko-coin-3`, since the exact method runs out of memory.

## 6.3 Results

`Sushi`. The agent with no auxiliary reward has no incentive to interfere with the sushi and goes directly to the goal. Since the starting state is unreachable no matter what the agent does, the reversibility reward is always 0, so it does not produce interference behavior. The future task agent with no baseline interferes with the sushi, while the agent with a baseline goes directly to the goal.

`Vase`. Since the agent can get to the goal without irreversible actions, both the reversibility and future task methods avoid the side effect on this environment, while the regular agent breaks the vase.

`Box`. The reversibility agent loses its auxiliary reward no matter how it moves the box, so it takes the fastest path to the goal that pushes the box in the corner (similarly to the regular agent). However, the future task agent pushes the box to the right, since some future tasks involve moving the box.

`Soko-coin`. Since the reversibility agent loses its auxiliary reward by collecting coins, it pushes boxes next to walls to get to the coins faster (similarly to the regular agent). However, the future task agent goes around the boxes to preserve future tasks that involve moving boxes.

The results are shown in Table 1. Each exact Q-learning agent converged to the optimal policy given by value iteration for the corresponding auxiliary reward. Only the future task approach with the baseline policy does well on all environments, avoiding side effects as well as interference. While the UVFA approximation of the future task auxiliary reward avoids side effects less reliably than the exact version, it shows some promise for scaling up the future task approach.

## 7   Other related work

**Side effects criteria using state features.** Minimax-regret querying [26] assumes a factored MDP where the agent is allowed to change some of the features and proposes a criterion for querying the supervisor about changing other features in order to allow for intended effects. RLSP [21] defines an auxiliary reward for avoiding side effects in terms of state features by assuming that the starting state

of the environment is already organized according to human preferences. While these approaches are promising, they require a set of state features in order to compute the auxiliary reward, which increases the burden on the reward designer.

**Empowerment.** The future task approach is related to *empowerment* [9, 18], a measure of the agent's control over its environment. Empowerment is defined as the maximal mutual information between the agent's actions and the future state, and thus measures the agent's ability to reliably reach many states. Maximizing empowerment would encourage the agent to avoid irreversible side effects, but would also incentivize interference, and it is unclear to us how to define an empowerment-based measure that would avoid this. One possibility is to penalize the reduction in empowerment between the current state $s_T$ and the baseline $s'_T$. However, empowerment is indifferent between these two cases: A) the same states are reachable from $s_T$ and $s'_T$, and B) a state $x$ is reachable from $s'_T$ but not from $s_T$, while another state $y$ is reachable from $s_T$ but not from $s'_T$. Thus, penalizing reduction in empowerment would miss some side effects: e.g. if the agent replaced the sushi on the conveyor belt with a vase, empowerment could remain the same, so the agent is not penalized for breaking the vase.

**Safe exploration.** While the safe exploration problem may seem similar to the side effects problem, safe exploration is about avoiding harmful actions during the training process (a learning problem), while the side effects problem is about removing the incentive to take harmful actions (a reward design problem). Many safe exploration methods work by changing the agent's incentives, and thus can potentially address the side effects problem. This includes reversibility methods [6], which avoid side effects in tasks that don't require irreversible actions. Safe exploration methods that penalize risk [4] or use intrinsic motivation [14] help the agent avoid side effects that result in lower reward (such as getting trapped or damaged), but do not discourage the agent from damaging the environment in ways that are not penalized by the reward function (e.g. breaking vases). Thus, safe exploration methods offer incomplete solutions to side effects, just as side effects methods provide incomplete solutions to safe exploration. These methods can be combined if desired to address both problems.

**Uncertainty about the objective.** Inverse Reward Design [8] incorporates uncertainty about the objective by considering alternative reward functions that are consistent with the given reward function in the training environment, and following a risk-averse policy. This helps avoid side effects that stem from distributional shift, where the agent encounters a new state that was not seen during training. However, along with avoiding harmful new states, the agent also avoids beneficial new states. Another uncertainty method is quantilization [23], which incorporates uncertainty by sampling from the top quantile of actions rather than taking the optimal action. This approach does not consistently remove the incentive for side effects, since harmful actions will still be sampled some of the time.

**Human oversight.** An alternative to specifying an auxiliary reward is to teach the agent to avoid side effects through human oversight, such as inverse reinforcement learning [17, 7], demonstrations [1], or human feedback [5, 19]. It is unclear how well an agent can learn a reward for avoiding side effects from human oversight. We expect this to depend on the diversity of settings in which it receives oversight and its ability to generalize from those settings, while an intrinsic reward for avoiding side effects would be more robust and reliable. Such an auxiliary reward could also be combined with human oversight to decrease the amount of human input required for an agent to learn human preferences, e.g. if used as a prior for the learned reward function.

# 8 Conclusions

To address the challenge of defining what side effects are, we have proposed a approach where a definition of side effects is automatically implied by the simpler definition of future goals, which lays a theoretical foundation for formalizing the side effects problem. This approach provides an auxiliary reward for preserving the ability to perform future tasks that incentivizes the agent to avoid side effects, whether or not the current task requires irreversible actions, and does not introduce interference incentives for the agent.

There are many possible directions for follow-up work, which include improving the UVFA approximation of the future task reward to more reliably avoid side effects, applying the method to more complex agents and environments, generalizing interference avoidance to the stochastic case, investigating the choice of future task distribution $F$ (e.g. incorporating human preferences by learning the task distribution through human feedback methods [5]), and investigating other possible undesirable incentives that could be introduced besides interference incentives.

## Broader Impact

In present-day reinforcement learning, what the agent should not do is usually specified manually, e.g. through constraints or negative rewards. This ad-hoc approach is unlikely to scale to more advanced AI systems in more complex environments. Ad-hoc specifications are usually incomplete, and more capable AI systems will be better at finding and exploiting gaps and loopholes in the specification. We already see many examples of specification gaming with present-day AI systems, and this problem is likely to get worse for more capable AI systems [11].

We think that building and deploying more advanced AI systems calls for general approaches and design principles for specifying agent objectives. Our paper makes progress on developing such a general principle, which aims to capture the heuristic of "do no harm" in terms of the available options in the environment, and gives the agent an incentive to consider the future consequences of its actions beyond the current task.

Without a reliable and principled way to avoid unnecessary changes to the world, the deployment of AI systems will be limited to narrow domains where the designer can enumerate everything the agent should not do. Thus, general approaches to objective specification would enable society to reap the benefits of applying capable AI systems to more difficult problems, which has potential for high long-term impact.

In terms of negative impacts, adding an auxiliary reward for future tasks increases the computational requirements and thus the energy cost of training reinforcement learning algorithms, compared to hand-designed rewards and constraints for avoiding side effects. The remaining gaps in the theoretical foundations of our method could lead to unexpected issues if they are not researched properly and instead left to empirical evaluation.

## Acknowledgments and Disclosure of Funding

We thank Ramana Kumar for detailed and constructive feedback on paper drafts and code. We also thank Jonathan Uesato, Matthew Rahtz, Alexander Turner, Carroll Wainwright, Stuart Armstrong, and Rohin Shah for helpful feedback on drafts.

This work was funded by Alphabet.

## Footnotes

[2]Code: `github.com/deepmind/deepmind-research/tree/master/side_effects_penalties`

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
