[Supplementary Material]

# A  Example 1

Consider a deterministic MDP with two states $x_0$ and $x_1$ and two actions a and b, where $x_0$ is the initial state. Suppose the baseline policy $\pi'$ always chooses action a.

We show that for most future task distributions, the future task auxiliary reward induces an interference incentive: staying in $x_0$ has a higher no-reward value than following the baseline policy.

The optimal value function $V_i^*$ for future task $i$ with goal state $x_i$ ($i \in \{0,1\}$) is $V_i^*(x_i) = 1$, $V_1^*(x_0) = \gamma$ and $V_0^*(x_1) = 0$. Then the no-reward value function for the baseline policy is

$$W_{\pi'}(x_0) = r_{\text{aux}}(x_0) + (\gamma + \gamma^2 + \dots) r_{\text{aux}}(x_1) = r_{\text{aux}}(x_0) + \frac{\gamma}{1-\gamma} \cdot (1-\gamma)\beta F(1) = r_{\text{aux}}(x_0) + \gamma \beta F(1)$$

and the no-reward value function for the policy $\pi_{\text{int}}$ that always takes action $b$ is

$$W_{\pi_{\text{int}}}(x_0) = r_{\text{aux}}(x_0) + \frac{\gamma}{1-\gamma} r_{\text{aux}}(x_0) = r_{\text{aux}}(x_0) + \gamma\beta(F(0) + \gamma F(1))$$

The future task agent has an interference incentive if the baseline policy is not optimal for the future task reward, i.e. $W_{\pi'}(x_0) < W_{\pi_{\text{int}}}(x_0)$. This happens iff $F(0) > (1-\gamma)F(1)$, i.e. if the task distribution does not highly favor task 1 over task 0.

# B  Proofs

## B.1  Proof of Proposition 1

**Proposition 1** (Value function convergence). The following formula for the optimal value function satisfies the goal condition $V_i^*(s_t, s_t') = r_i(s_t, s_t') = 1$ and Bellman equation (4):

$$V_i^*(s_t, s_t') = \mathbb{E}\left[\gamma^{\max(N_i(s_t), N_i(s_t'))}\right] = \sum_{n=0}^{\infty} \mathbb{P}(N_i(s_t) = n) \sum_{n'=0}^{\infty} \mathbb{P}(N_i(s_t') = n')\gamma^{\max(n,n')} \quad (5)$$

*Proof. Goal condition.* If both agents have reached the goal state ($s_t = s_t' = g_i$), then $N_i(s_t) = N_i(s_t') = 0$, so formula (5) gives $V_i^*(s_t, s_t') = 1$ as desired.

*Bellman equation.* We show that formula (5) is a fixed point for the Bellman equation (4).

If $s_t \neq g_i$, and thus $\mathbb{P}(N_i(s_t) = 0) = 0$, we note that

$$\sum_{m=0}^{\infty} \mathbb{P}(N_i(s_t) = m)\gamma^{\max(m,k)} = \sum_{n=0}^{\infty} \mathbb{P}(N_i(s_t) = n+1)\gamma^{\max(n+1,k)}$$

$$= \max_{a_t \in \mathcal{A}} \sum_{s_{t+1}} p(s_{t+1}|s_t, a_t) \sum_{n=0}^{\infty} \mathbb{P}(N_i(s_{t+1}) = n)\gamma^{\max(n+1,k)} \quad (6)$$

if we decompose the trajectory to the goal state into the first transition $(s_t, a_t, s_{t+1})$ and the rest of the trajectory starting from $s_{t+1}$. This holds analogously for the reference transition $(s_t', a_t', s_{t+1}')$.

Now we plug in formula (5) on the right side (RS) of the Bellman equation. If $s_t \neq g_i$ and $s_t' \neq g_i$:

$$RS = r_i(s_t, s_t') + \gamma \max_{a_t \in \mathcal{A}} \sum_{s_{t+1} \in \mathcal{S}} p(s_{t+1}|s_t, a_t) \sum_{s_{t+1}' \in \mathcal{S}} p(s_{t+1}'|s_t', a_t')V_i^*(s_{t+1}, s_{t+1}')$$

$$= 0 + \gamma \max_{a_t \in \mathcal{A}} \sum_{s_{t+1} \in \mathcal{S}} p(s_{t+1}|s_t, a_t) \sum_{s_{t+1}' \in \mathcal{S}} p(s_{t+1}'|s_t', a_t') \cdot$$

$$\sum_{n=0}^{\infty} \mathbb{P}(N_i(s_{t+1}) = n) \sum_{n'=0}^{\infty} \mathbb{P}(N_i(s_{t+1}') = n')\gamma^{\max(n,n')} \quad \text{[plugging in (5)]}$$

$$= \max_{a_t \in \mathcal{A}} \sum_{s_{t+1} \in \mathcal{S}} p(s_{t+1}|s_t, a_t) \sum_{n=0}^{\infty} \mathbb{P}(N_i(s_{t+1}) = n) \cdot \quad \text{[moving } \gamma, \text{ rearranging sums]}$$

$$\sum_{s'_{t+1} \in \mathcal{S}} p(s'_{t+1}|s'_t, a'_t) \sum_{n'=0}^{\infty} \mathbb{P}(N_i(s'_{t+1}) = n') \gamma^{\max(n+1, n'+1)}$$

$$= \max_{a_t \in \mathcal{A}} \sum_{s_{t+1} \in \mathcal{S}} p(s_{t+1}|s_t, a_t) \sum_{n=0}^{\infty} \mathbb{P}(N_i(s_{t+1}) = n) \cdot$$

$$\sum_{m'=0}^{\infty} \mathbb{P}(N_i(s'_t) = m') \gamma^{\max(n+1, m')} \quad \text{[using (6) with } m' = n' + 1]$$

$$= \sum_{m=0}^{\infty} \mathbb{P}(N_i(s_t) = m) \sum_{m'=0}^{\infty} \mathbb{P}(N_i(s'_t) = m') \gamma^{\max(m, m')} \quad \text{[using (6) with } m = n + 1]$$

which is the same as plugging in formula (5) on the left side.

If $s'_t = g_i$, we have:

$$RS = r_i(s_t, g_i) + \gamma \max_{a_t \in \mathcal{A}} \sum_{s_{t+1} \in \mathcal{S}} p(s_{t+1}|s_t, a_t) V_i^*(s_{t+1}, g_i)$$

$$= 0 + \gamma \max_{a_t \in \mathcal{A}} \sum_{s_{t+1} \in \mathcal{S}} p(s_{t+1}|s_t, a_t) \sum_{n=0}^{\infty} \mathbb{P}(N_i(s_{t+1}) = n) \gamma^{\max(n, 0)} \quad \text{[plugging in (5)]}$$

$$= \sum_{m=0}^{\infty} \mathbb{P}(N_i(s_{t+1}) = m) \gamma^{\max(m, 0)} \quad \text{[using (6) with } m = n + 1]$$

If $s_t = g_i$, we have:

$$RS = r_i(g_i, s'_t) + \gamma \sum_{s_{t+1} \in \mathcal{S}} p(s_{t+1}|s'_t, a'_t) V_i^*(g_i, s'_{t+1})$$

$$= 0 + \gamma \sum_{s'_{t+1} \in \mathcal{S}} p(s'_{t+1}|s'_t, a'_t) \sum_{n'=0}^{\infty} \mathbb{P}(N_i(s'_{t+1}) = n') \gamma^{\max(0, n')} \quad \text{[plugging in (5)]}$$

$$= \sum_{m'=0}^{\infty} \mathbb{P}(N_i(s'_{t+1}) = m') \gamma^{\max(0, m')} \quad \text{[using (6) with } m' = n' + 1]$$

which is the same as plugging in formula (5) on the left side. $\square$

## B.2 Proof of Proposition 2

**Proposition 2** (Avoiding interference). *For any policy $\pi$ in a deterministic environment, the baseline policy $\pi'$ has the same or higher no-reward value: $W_\pi(s_0) \leq W_{\pi'}(s_0)$.*

*Proof.* Suppose policy $\pi$ is in state $s_k$ at time $k$. Then,

$$W_\pi(s_0) = \sum_{T=0}^{\infty} \gamma^T r_{\text{aux}}(s_k)$$

$$= \sum_{T=0}^{\infty} \gamma^T (1 - \gamma) \beta \sum_i F(i) V_i^*(s_T, s'_T)$$

$$= \sum_{T=0}^{\infty} \gamma^T (1 - \gamma) \beta \sum_i F(i) \gamma^{\max(N_i(s_T), N_i(s'_T))}$$

$$\leq \sum_T \gamma^T (1-\gamma)\beta \sum_i F(i) \gamma^{N_i(s'_T)}$$

$$= \sum_{T=0}^{\infty} \gamma^T (1-\gamma)\beta \sum_i F(i) V_i^*(s'_T, s'_T)$$

$$= \sum_{T=0}^{\infty} \gamma^T r_{\text{aux}}(s'_T)$$

$$= W_{\pi'}(s_0)$$

$\square$

## C  Interference in the stochastic case

If the environment is stochastic, the outcome of running the baseline policy varies. For example, suppose the agent is in a room, following a baseline policy of doing nothing (staying in one location in the room). A human walks into the room, and 10% of the time they knock over a vase. We want the agent to avoid interfering with the human in those 10% of cases, and also to avoid breaking the vase the other 90% of the time. This indicates that we want to compare the agent's effects to a specific counterfactual (either the human was going to walk in or not) rather than an average of all possible counterfactuals sampled from the stochastic environment (10% probability of the human walking in). Thus, we would like the baseline policy to be optimal for any deterministic instantiation of the stochastic environment (e.g. by conditioning on the random seed). Then the auxiliary reward, which is a function of the baseline state, will depend on the random seed.

Thus, the definition of interference could be refined as follows: there is an interference incentive if the baseline policy is not optimal from the initial state, conditioning on the random seed of the environment. However, it is unclear how this could be implemented in practice outside simulated environments.

## D  Stepwise application of the baseline policy

### D.1  Inaction rollouts

Figure D.1: Inaction rollouts from the current state $s_T$ and baseline state $s'_T$, obtained by applying the baseline policy to those states: $s_{\tilde{T}+1} = \pi'(s_T)$, etc. If the previous action $a_{T-1}$ drops the vase from the building, then the vase breaks in the inaction rollout from $s_T$ but not in the inaction rollout from $s'_T$.

### D.2  Offsetting incentives

Unlike the initial mode, the stepwise mode avoids incentives for *offsetting* behavior, where the agent undoes its own actions towards the objective [10]. For example, consider a variant of the Sushi environment without a goal state, where the object on the belt is a vase that falls off and breaks if it reaches the end of the belt, and agent receives a reward for taking the vase off the belt.

Then the initial mode gives the agent an incentive to take the vase off the belt (collecting the reward) and then offset this action by putting the vase back on the belt. This type of offsetting is undesirable,