[Reviews · NeurIPS 2020]

Review 1

Summary and Contributions: The authors have addressed my main concerns and confusions and I think the changes noted in the rebuttal will make this a stronger paper. The fixes to the confusing notation and the clarification of the algorithmic details are appreciated and will make the paper easier to understand. --- This paper seeks to address the problem of side effects: where an agent changes an environment in unnecessary or undesirable ways when seeking to accomplish a task. The authors propose that side effects can be avoided if an agent thinks not only about its current task but also future tasks that it may later need to accomplish. The authors also introduce a baseline policy to provide an incentive to not interfere.

Strengths: This paper proposes a nice general way to address side effects. The idea of thinking about the future seems like a nice elegant way to address this issue. Empirical experiments show that agents whose policies are optimized via the proposed framework have fewer side effects than other approaches.

Weaknesses: The beginning of the paper was well written and easily accessible. However, the notation quickly becomes overloaded and it is difficult to understand how section 4 connects to the previous sections which are better explained. In particular, there is not a clear algorithm that is defined and it is difficult to determine exactly what the proposed method is. Section 2 defines the future task framework nicely, but then Section 3 shows that there can be problems with interference. Section 4 purports to solve these problems but it is never made clear exactly what needs to change from section 2. The proposed approach also only works with binary goal-based rewards for deterministic environments. Furthermore, the proposed approach requires an optimal reference policy for any possible subgoal. This seems highly restrictive since the goal seems to be to assist people in specifying reward functions, but if the optimal policy for any subgoal is already known, then why do we care about reward specification? If we can assume to have optimal policies for all goals, we probably also have an optimal policy that avoids side effects, etc.

Correctness: Definition 3 references a baseline \pi' but doesn't include it in the equation. This seems incorrect. The experimental methodology seems sounds.

Clarity: The notation is confusing wrt V. The subscripts and superscripts are overloaded and sometimes seems inconsistent. For example, based on previous definitions, V_i^*(s) is the optimal value function for task i starting at state s, but in Example 1, we have V_0^*(s_1) = \gamma which seems to make the subscript the start state and the argument the goal state. Finally, as noted earlier, Section 4 presents nice theory but it is unclear exactly how this section should be combined with section 2 for the full approach. Having a nice summary or algorithm box would be very helpful to make the approach clear and enable better reproducibility.

Relation to Prior Work: The related work section does a good job of relating the work to existing research.

Reproducibility: No

Additional Feedback: How are goal states sampled for the experiments? Do you enforce that all goal states are actually reachable? Consider the the egg example, it seems like the agent will never use the eggs to make breakfast if it thinks it might need the eggs to make a cake later. There is always a posibliity of needed a limited resource so will the agent will reach analysis paralysis and not be able to do anything for fear of missing out on something in the future? Line 115: Last sentence is ambiguous about what "this" refers to. How do you pick the baseline? What is a good baseline. Doing nothing as an autonomous car seems bad since you won't get anywhere. In many settings doing nothing seems like it could be pessimal.


Review 2

Summary and Contributions: In the search of good rewards by punishing undesirable side-effects, such as breaking a vase, while reaching a goal state, such as reaching the kitchen from the hallway. Rather than defining the undesirable side-effect as an irreversible action, as proposed in literature, the authors propose an auxiliary reward which incorporates the possibility to perform a next task, such as filling the vase with flowers. This auxiliary reward takes the form of the value estimate at the terminal state of the current task for predicting the discounted future return for the potential next task.

Strengths: Relevant problem, with elegant solution. Thorough derivation. Proper description of the problem. Novel method.

Weaknesses: What about more complex rewards? Does the proof still hold? The proof accounts for a very simplistic reward: 1 @ goal state and 0 otherwise. Is a grid world evaluation common for this type of research, or should their be more complex environments? What is the impact of the bias/variance trade-off: extra variance introduced making potentially the training more difficult? Benchmarks would be welcome. Irreversible auxiliary reward is discarded by one simple negative results. However, I could think of a grid-world scenario with too many side-effects which are not all included by future goals, and therefore not included in this future-task auxiliary reward. For example, walking in a museum would require future tasks for each statue, in order to train the robot to avoid them. This unknown set of potential outcomes is more easily covered with irreversibility.

Correctness: The limitations of the proposed method in comparison to the related work is probably not clearly enough stated, leading to an oversell of the method.

Clarity: Well written. Clear explanation of the method.

Relation to Prior Work: The prior work section appears complete and is well compared with the proposed new method. The paper proposes a more thorough definition on side-effects. In the section on safe exploration, it appears that safe-exploration is insufficient. However, the same could be said about the side-effect problem: A damaged agent might be able to perform future tasks but is still a damaged agent. Risk-aversity is implied in the side-effect problem indirectly, however, this might be insufficient for more risky tasks.

Reproducibility: Yes

Additional Feedback: The paper starts with ‘designing rewards is difficult’ for the designer, however, the paper does not really mention any reward shaping solutions. This sets the reader on the wrong track. I would recommend something more according to these lines: avoiding undesired effects on the environment from a goal-reaching agent is challenging as side effects are hard to incorporate in a reward without giving the wrong incentive. A big challenge comes from the difficulty of encountering unforeseen side-effects. This difficulty is more broadly countered by the reversibility auxiliary reward function. Related to the conclusions on future work: By incorporating one future task in the value estimate, the same idea could further be explored in a hierarchical RL setting where likelihood over tasks are taken into account which is something we, as humans, do constantly: I’m not cleaning the cutting plank just yet as I might use it in its current state for cutting my next vegetables.


Review 3

Summary and Contributions: We are typically unable to specify exactly what we would like in a handcoded reward function for reasonably complex tasks. In particular, it is hard to specify all the side effects -- all of the things that the agent should *not* change about the environment. So, we would like a generic method that can penalize side effects in arbitrary environments for an arbitrary reward function. A simple approach is to have our agent preserve its options, which the authors formalize by having the agent maintain its ability to pursue a distribution of future tasks. However, this leads to interference incentives -- if something were going to restrict the agent’s option value, such as a human irreversibly eating some food, the agent would be incentivized to interfere with that process in order to keep its option value for the future. The authors provide a formal definition of this incentive. To fix this problem, the authors introduce a baseline policy (which could be set to e.g. noop actions), and propose a future task reward that only provides reward if the baseline policy would have been able to complete the future task. Thus, the agent is only incentivized to preserve options that “would have been available”. The authors show via experiments with simple gridworlds that the future task approach with the baseline allows us to avoid side effects, while also not having intervention incentives.

Strengths: The problem is important: reward specification is hard to get right, and techniques to ameliorate the difficulty are important and understudied. I particularly like the conceptual analysis of what is necessary for avoiding both side effects and intervention incentives: the reasoning is compelling, and the conclusions are borne out by the empirical results.

Weaknesses: The experiments are done on relatively simple gridworlds, so it is not easy to say whether the method will work in more complex environments. Nonetheless, I see the contribution as the conceptual analysis, so this is not a major point. See also the relation to prior work section below.

Correctness: To my knowledge, yes.

Clarity: Yes.

Relation to Prior Work: Previous work on relative reachability (that the authors cite) has introduced the idea of a baseline policy and intervention incentives, including a stepwise inaction baseline that cannot be represented in the formalism of this paper because it depends on the agent’s policy (whereas the future tasks framework requires a baseline policy that is independent of the agent’s policy). This is discussed in Appendix A.2 -- I might recommend that the authors move some of this discussion into the main paper.

Reproducibility: Yes

Additional Feedback: After reading the author response and the other reviews, I am keeping my score. I did not raise substantial critiques, and I found the authors’ response convincing at rebutting the critiques of the other reviewers.

[Author Response · NeurIPS 2020]

**Common** We thank the reviewers for their thorough and helpful feedback. The reviewers noted the importance of the
problem and the generality and elegance of the proposed solution. We address their concerns below, and incorporate all
our clarifications into the paper.
**Binary goal-based future task rewards.** Note that we do *not* make this assumption for the *current* task reward
function, which can be arbitrary. We think that simple future task rewards of this form are sufficient to cover a wide
variety of future goals and thus effectively penalize side effects. More complex future tasks can often be decomposed
into such simple tasks, e.g. if the agent preserves the ability to move boxes 1 and 2 in Soko-coin, then it can also
perform a task involving both boxes. Regarding proofs, the value function formula given in Proposition 1 only applies
to goal-based future task rewards (otherwise the goal distance is not defined), while the formula for the general case
with arbitrary future task rewards is much messier. Binary goal-based rewards allow us to cover the space of future
tasks while keeping the theory simple, so we do not consider this assumption a significant limitation.
**Gridworlds.** Gridworld evaluation is common in this research area, e.g. in references [20] (ICLR 2019), [23] (AIES
2020), [24] (IJCAI 2018). We agree with the importance of evaluating on more complex environments in future work.

**R1** **Section 4 clarity.** We add an algorithm (Figure 1) to
illustrate the proposed approach. The algorithm for Section 2
can be obtained by removing lines 1, 7, 11, and setting line 5
to "if $s_t = g_i$: set $\mathbf{r_i(s_t)} := \mathbf{1}$, break" (marked in blue).
**Section 3 typos.** In Definition 3, the reference to the baseline
policy should be omitted, and $V_\pi(s_{t+1})$ should be $V_\pi^0(s_{t+1})$.
In Example 1, the $V_i^*$ formulas are backwards - they should be
$V_0^*(s_1) = 0$ and $V_1^*(s_0) = \gamma$. We thank R1 for pointing out
these errors in the writeup, and apologize for the confusion.
**Optimal policies.** There may be a misunderstanding here -
our assumption is that we can find an (approximately) optimal
policy for any subgoal with a well-defined reward function
(such as reaching a single goal state). This is orthogonal to the
reward specification problem we highlight - that it is difficult
to define a good reward function in the first place. The future

Figure 1: Algorithm for Section 4

1: Set $s_0' := s_0$
2: **for** $T = 0$ **to** $T_{max}$ **do**
3:     Draw task $i \sim F$
4:     **for** $t = T$ **to** $T + T_{max}$ **do**
5:         if $s_t = s_t' = g_i$: set $\mathbf{r_i(s_t, s_t')} := \mathbf{1}$, break
6:         if $s_t \neq g_i$: $a_t \sim \pi_i(s_t)$, $s_{t+1} \sim p(s_t, a_t)$
7:         if $s_t' \neq g_i$: $a_t'^* \sim \pi_i^*(s_t')$, $s_{t+1}' \sim p(s_t', a_t'^*)$
8:     **end for**
9:     if $s_T$ is terminal: break
10:     $a_T \sim \pi(s_T)$, $s_{T+1} \sim p(s_T, a_T)$
11:     $a_T' \sim \pi'(s_T')$, $s_{T+1}' \sim p(s_T', a_T')$
12: **end for**

tasks framework addresses this problem by allowing us to define a reward function for the goal of avoiding side effects.
Note that in practice our method does not require actually running an optimal policy for every subgoal - we only need
to compute the value function formula in Proposition 1, which can be approximated using UVFA.
**Notation.** We can reduce notation overloading for value functions by renaming $V_\pi^0$ to $W^\pi$.
**Goal states.** (Reachability) We do not enforce that goal states are reachable. If a goal is unreachable, then the agent
gets no reward for the corresponding future task, and is not penalized since the reference agent also cannot reach this
goal. (Sampling) We sample 10 goal states from a stored set of 100 states encountered by the agent. Whenever a state is
encountered that is not in the stored set, it randomly replaces another state in the stored set with probability 0.01.
**Egg example.** For high values of $\beta$, it's true that the agent will avoid using the egg in the current task because it may
be needed in future tasks. The value of $\beta$ needs to be set low enough for the agent to succeed at the current task.
**Baseline.** We agree that it's often not obvious how to choose the baseline policy. Note that the baseline policy represents
what happens by default, rather than a safe course of action or an effective strategy for achieving a goal (so the baseline
is task-independent). In the car example, the default outcome for the car is sitting in the garage and not getting anywhere.
It is certainly true that doing nothing can have bad outcomes (e.g. standing by while a human driver causes a collision).
However, since the agent does not cause these outcomes, they don't count as side effects of the agent's actions. The role
of the baseline policy is to filter out these outcomes that are not caused by the agent (to avoid interference incentives).

**R2** **Comparison to reversibility.** We agree that the reversibility reward covers unforeseen side effects (as long as
they make the initial state harder to reach). However, as noted in the introduction, the reversibility reward is not sensitive
to the magnitude of these side effects, and thus would not penalize setting the kitchen on fire relative to breaking an egg
(since both are irreversible). Note that the future task approach incorporates the reversibility reward as a future task $i$
whose goal state is the initial state ($g_i = s_0$), since the future tasks are sampled uniformly from all possible goal states.
Thus, the future task approach covers all the side effects that are covered by the reversibility reward.
**Comparison to safe exploration.** We agree that just as safe exploration methods are insufficient to address the side
effect problem, side effects methods like future tasks are not sufficient to address the safe exploration problem. We
would recommend combining these methods in order to address both problems.
**Bias-variance trade-off.** We assume you are referring to the variance introduced by using an auxiliary reward function
that changes as it is being learned during training. We agree that this probably makes the training process more unstable,
and may be responsible for some of the variance in the experimental results. We have also tried freezing the auxiliary
reward function after an initial period of exploration, but found that this did not reduce the variance in the results.

**R3** **Stepwise baseline.** Agreed. We moved the proofs to the appendix and moved this discussion to the main paper.

[Meta-Review · NeurIPS 2020]

Some reviewers were initially concerned about the restriction of evaluation to grid worlds and requiring simplistic rewards and determinism. However, the author response clarified and proposed changes that satisfied the reviewers, who came to consensus that the paper should be accepted.